# Transient Waterlogging Events Impair Shoot and Root Physiology and Reduce Grain Yield of Durum Wheat Cultivars

**DOI:** 10.3390/plants10112357

**Published:** 2021-11-01

**Authors:** Lorenzo Cotrozzi, Giacomo Lorenzini, Cristina Nali, Claudia Pisuttu, Silvia Pampana, Elisa Pellegrini

**Affiliations:** 1Department of Agriculture, Food and Environment, University of Pisa, Via del Borghetto 80, 56124 Pisa, Italy; lorenzo.cotrozzi@agr.unipi.it (L.C.); giacomo.lorenzini@unipi.it (G.L.); cristina.nali@unipi.it (C.N.); claudia.pisuttu@phd.unipi.it (C.P.); elisa.pellegrini@unipi.it (E.P.); 2CIRSEC, Centre for Climate Change Impact, University of Pisa, Via del Borghetto 80, 56124 Pisa, Italy

**Keywords:** abiotic stress, antioxidants, climate change, flooding, osmoprotectans, reactive oxygen species, *Triticum turgidum* L. subsp. *durum*, yield

## Abstract

Durum wheat (*Triticum turgidum* L. subsp. *durum* (Desf.) Husn) is a staple crop of the Mediterranean countries, where more frequent waterlogging events are predicted due to climate change. However, few investigations have been conducted on the physiological and agronomic responses of this crop to waterlogging. The present study provides a comprehensive evaluation of the effects of two waterlogging durations (i.e., 14 and 35 days) on two durum wheat cultivars (i.e., Svevo and Emilio Lepido). An integrated analysis of an array of physiological, biochemical, biometric, and yield parameters was performed at the end of the waterlogging events, during recovery, and at physiological maturity. Results established that effects on durum wheat varied depending on waterlogging duration. This stress imposed at tillering impaired photosynthetic activity of leaves and determined oxidative injury of the roots. The physiological damages could not be fully recovered, subsequently slowing down tiller formation and crop growth, and depressing the final grain yield. Furthermore, differences in waterlogging tolerance between cultivars were discovered. Our results demonstrate that in durum wheat, the energy maintenance, the cytosolic ion homeostasis, and the ROS control and detoxification can be useful physiological and biochemical parameters to consider for the waterlogging tolerance of genotypes, with regard to sustaining biomass production and grain yield.

## 1. Introduction

Durum wheat (*Triticum turgidum* L. *subsp. durum* (Desf.) Husn) is one of the oldest cultivated cereals and plays a pivotal role in global food security. Although the necessity of wheat grain will increase by 60%, its production might decline by 29% because of the environmental stresses driven by climate change [1]. Durum wheat is among the most widespread and economically important crops in the Mediterranean countries [2], notwithstanding the Mediterranean environment, which is recognized to be extremely vulnerable to climatic changes [3]. Specifically, it has been predicted that durum wheat will be affected by more recurrent, severe, and unpredictable flooding events [4].

In rain-fed situations, flooding happens when more rain falls than the soil can absorb, or the atmosphere can evaporate. In central Italy, excess water is likely to occur from October to April but is more expected during the winter months (from January to February), due to lower transpiration and evaporation rates of the crop. Therefore, durum wheat is more prone to excess water during the tillering stage, which is critical for tiller production and spikelet initiation [5]. Based on the height of the water column produced, flooding can be classified as waterlogging when water covers just the root system, or as submergence, when water also overlays the plant’s aerial organs [6]. In waterlogged soils, gas diffusion through soil pores is inhibited, so the oxygen (O_2_) concentration decreases rapidly, while the carbon dioxide (CO_2_) and ethylene concentrations increase in the root environment [7]. A slowed O_2_ influx is the main cause of injury to the roots and to the shoots they support [8]. The plants react through a series of morphological and physiological responses to the damages due to O_2_ deprivation. From a physiological point of view, excess water accumulation in the root zone can induce osmotic stress and disrupt cell ion homeostasis. To cope with such stressful conditions, plants tend to accumulate compounds called osmoprotectans (such as free amino acids, non-structural carbohydrates, and quaternary ammonium substances), as their accumulation may decrease the osmotic potential [9,10]. Next, the impaired root functioning under waterlogging affects the physiological responses of the shoots, particularly the carbon fixation. Waterlogging may induce partial stomatal closure that, in turn, could constrain internal CO_2_ levels and limit carbon fixation [11,12]. Additionally, photosynthesis rates can also be inhibited by non-stomatal factors, e.g., oxidative injury [10] caused by reactive oxygen species (ROS) like hydrogen peroxide (H_2_O_2_), superoxide radicals, and hydrogen radicals which impair mesophyll conductance [13], and harm photosystem II (PSII), causing cellular damage and leaf chlorosis related to chlorophyll degradation [14,15].

In durum wheat, O_2_ deficiency caused by waterlogging has been demonstrated to prematurely induce leaf senescence, reduce root and shoot growth, and constrain spike development, thus decreasing the final grain yield of the crop [5]. However, the effect of transient waterlogging could be somewhat compensated by the subsequent recovery of the growth of roots and shoots, as demonstrated in other winter cereals, such as oat [16] and barley [17]. Recovery involves the allocation of carbon to roots after waterlogging and hypoxia for preferential root growth, to re-establish a root-to-shoot ratio typical of plants with drained soils. This preferential resource allocation to root growth would be a major reason explaining the reduced shoot growth following a period of waterlogging [18]. Nevertheless, to the best of our knowledge, very little research has addressed the effects of waterlogging throughout the entire crop cycle, describing both vegetative growth and grain production. Thus, evidence on root and shoot growth during waterlogging and subsequent recovery is limited to oat, barley, and common wheat [16,17], with no confirmation existing for durum wheat.

To fill the gap of knowledge about the mechanism(s) of the response of durum wheat to waterlogging and to relate the physiological responses of leaves and roots to the growth, recovery ability, and final grain yield of the crop, the present research aimed to investigate the effects of different waterlogging durations (i.e., 14 and 35 days) at tillering on the growth and the grain yield of durum wheat, as well as identify the main physiological traits involved in the response of roots, shoots, and leaves.

As the cultivar choice may represent a key factor in coping with waterlogging [5,19], we compared the two durum wheat genotypes Svevo and Emilio Lepido. To the best of our knowledge, the waterlogging tolerance of durum wheat cultivars currently cultivated in Italy has been previously studied only for Claudio and Svevo, which displayed very similar responses, as well [5]. In common wheat, previous research showed that high yielding genotypes were more affected by waterlogging [20]. Thus, for the present research, we selected two cultivars from those most cultivated in central Italy differing in cycle length and yielding capacity, assuming that Svevo could be less tolerant to waterlogging than Emilio Lepido due to its higher yielding capacity. The waterlogging durations were chosen because in previous experiments, we found that winter cereals exhibited grain yield reductions when waterlogging at tillering lasted for more than 16 days (barley) and 20 days (wheat and durum wheat) [5,19,21].

More specifically, our objectives were to assess the mechanism of response of the two durum wheat genotypes to 14 and 35 days of waterlogging at tillering, evaluating: (i) the immediate impairment of root and shoot growth and related physiological and biochemical parameters, as well as water status; (ii) the ability to recover from the end of waterlogging up to maturity; and (iii) the final grain yield.

## 2. Results

### 2.1. Meteorological Conditions

During the experiment (i.e., durum wheat cycle), the total rainfall was 672 mm spread over about 80 rainy days and was mainly concentrated in the period from December to February (Figure 1), as is typical of the autumn and spring growing season in central Italy. Temperatures ranged from 3.1 °C to 33.9 °C (recorded in February and June, respectively), and the daily mean temperature was 10.4 °C during the waterlogging imposition and 13.1 °C along the entire crop cycle, matching rather well with the historical data (1995–2020) for the site (13.0 °C).

### 2.2. Plant Phenology

Emilio Lepido started tillering at the same time of Svevo; the two cultivars also reached flowering simultaneously (Appendix A). Waterlogging slowed plant development, and plants of both cultivars waterlogged for the longest period (i.e., 35 days) reached flowering approximately one week later than the controls. However, waterlogged and control plants of both cultivars achieved maturity concurrently.

### 2.3. Waterlogging Immediate Effects on Physiological, Biochemical, and Biometric Parameters

Table 1 shows the effects of cultivar, waterlogging, and their interaction on leaf and root parameters evaluated at 0, 14, and 35 days of waterlogging (DOW). At the beginning of the experiment (i.e., 0 DOW), relative water content (RWC), leaf total chlorophyll (Chl_TOT_) and calcium ion (Ca^2+^), leaf and root malondialdehyde (MDA), and shoot-to-root biomass ratio values were higher in Emilio Lepido than in Svevo, while the maximum quantum efficiency of the photosystem II (PSII) photochemistry (F_v_/F_m_), chlorophyll a/b ratio (Chl a/b), de-epoxidation state (DEPS), leaf and root hydrogen peroxide (H_2_O_2_), leaf potassium ion (K^+^), root Ca^2+^, and shoot and root biomass levels were higher in Svevo (*data not shown*). Fourteen DOW reduced the CO_2_ assimilation rate (A) and stomatal conductance (g_s_) only in Svevo (−53 and −55%, respectively; throughout the whole text, percentages of waterlogging effects are calculated in comparison with the related controls), whereas 35 DOW reduced these parameters regardless of the cultivar (around −50%; Figure 2a,b). Intrinsic water-use efficiency (WUE_in_) increased only in Svevo at 35 DOW (+35%; Figure 2c).

The PSII operating efficiency in light conditions (Φ_PSII_) only decreased in Emilio Lepido at 35 DOW (−23%; Figure 2d). Differently, photochemical quenching (qP) was equally reduced by both 14 and 35 DOW in both cultivars (−6%; Figure 2e), whereas non-photochemical quenching (qNP) was increased in both cultivars at 14 DOW (around +40%) and only in Emilio Lepido at 35 DOW (+40%; Figure 2f).

Leaf osmotic potential (Ψ_π_) was reduced by 35 DOW (−17%; Figure 3a). RWC was reduced by 7% at 14 DOW and increased by 5% at 35 DOW (Figure 3b). Total chlorophyll content was reduced by 14 DOW only in Svevo (−28%), whereas 35 DOW decreased Chl_TOT_ in both cultivars, more in Emilio Lepido than in Svevo (−47 and −31%, respectively; Figure 4a). Leaf chlorosis was also visible with the naked eye.

Total carotenoids (Car_TOT_) were similarly reduced in both cultivars by 14 DOW (−27%) and even more by 35 DOW (−41%; Figure 4b).

MDA and H_2_O_2_ accumulations were not observed in the leaves of either cultivars; instead, leaf MDA levels were almost halved by both 14 and 35 DOW, and leaf H_2_O_2_ production was reduced by around 20% by 35 DOW (*data not shown*). Conversely, root MDA levels were increased by 14 DOW in both cultivars (almost doubled; Figure 5a), and root H_2_O_2_ content was noticeably increased by 14 DOW only in Emilio Lepido (more than five-fold) and by 35 DOW in both cultivars (more than two-fold in Emilio Lepido and +63% in Svevo; Figure 5b).

Leaf K^+^ content decreased by 14 DOW (−34% in both cultivars), whereas it was more decreased in Emilio Lepido than in Svevo by 35 DOW (−50 and −14%, respectively; Figure 6a). Leaf Ca^2+^ content increased only in Svevo at 14 DOW (+14%) and only in Emilio Lepido at 35 DOW (+47%; Figure 6b). Root K^+^ content was reduced by 14 DOW only in Svevo (−23%), whereas it was similarly reduced by 35 DOW in both cultivars (−45%; Figure 6c). A reduction in root Ca^2+^ was observed only in Emilio Lepido at 35 DOW (−47%; Figure 6d).

The number of culms per plant was reduced in both cultivars by 14 DOW (−18 and −21% in Svevo and in Emilio Lepido, respectively), whereas it decreased only in Svevo with 35 DOW (−60%; Figure 7a). Shoot biomass was reduced in both cultivars by 14 DOW (−27%), whereas it was decreased by 35 DOW (−91% in Svevo and −33% in Emilio Lepido; Figure 7b). Conversely, although root biomass was also reduced at both 14 and 35 DOW (−62 and −86%, respectively), no differential waterlogging effects were observed between cultivars (Figure 7c). The shoot-to-root biomass ratio increased in both cultivars due to 14 DOW (almost doubled), whereas it increased only in Emilio Lepido due to 35 DOW (more than three-fold; Figure 7d).

The canonical discriminant analysis gave seven significant new canonical variables (Can; *p* ≤ 0.001). Among the Cans, the first two (i.e., Can1 and Can2) accounted for 90.5% of the total variability (Appendix A), thus indicating that the multivariate structure of the original variables (i.e., all the above-reported parameters collected at the end of waterlogging events) can be well represented by these Cans. All experimental groups were discriminated, except for control and waterlogged Svevo plants at 14 DOW (Figure 8). Can1 mostly discriminated waterlogged plants of Emilio Lepido (exposed to both 14 and 35 DOW) from the others, especially from control plants of the same cultivar at the first time of analysis. Can1 was strongly and positively correlated with qNP and root H_2_O_2_, while it was strongly and negatively correlated with Car_TOT_ and leaf K^+^. Can2 mostly discriminated Svevo plants exposed to WL35 from the others, and it was strongly and positively correlated with WUE_in_ and strongly and negatively correlated with the number of culms.

### 2.4. Waterlogging Effects during Recovery at Physiological Level

Table 2 shows the effects of cultivar, waterlogging, and their interaction on physiological and water status parameters, as collected during the recovery period (i.e., 70 days from the beginning of waterlogging). No detrimental effects due to waterlogging were reported on gas exchange and chlorophyll *a* fluorescence. Conversely, A increased in Emilio Lepido subjected to waterlogging for 14 days (WL14, +53%) and g_s_ increased in both cultivars subjected to both WL14 and WL35, by about 35%. C_i_ increased in both cultivars subjected only with WL35 (+7%), Φ_PSII_ and qP were higher in Emilio Lepido subjected to WL35 (+14 and +11%), and qNP was lower in Emilio Lepido subjected to WL35 (−26%), as well as in Svevo subjected to both WL14 and WL35 (−24% and −37%, respectively). Nevertheless, similarly between cultivars, WUE_in_ was lower in plants subjected to WL35 (−18%).

### 2.5. Waterlogging Long-Lasting Effects on Final Grain Yield

Table 3 shows the effects of cultivar, waterlogging, and their interaction on biometric and yield parameters collected at maturity (i.e., 125 days from the beginning of waterlogging, lasting 14 or 35 days). Both Emilio Lepido and Svevo plants that had previously been subjected to WL35 showed a reduced number of culms (−29%), whereas no effects were observed on the number of spikes. Grain yield reduction was shown only in Svevo (−45 and −64% due to WL14 and WL35, respectively). The vegetative above-ground part was reduced by both WL14 and WL35 (−31 and −44%, respectively) without differences between the two cultivars, whereas the root biomass was reduced only by WL35 (−33% for Svevo and −42% for Emilio Lepido).

## 3. Discussion

Few studies have evaluated the impact of waterlogging on durum wheat [5,22], and to the best of our knowledge, there is no comprehensive research on the impact of waterlogging throughout the entire crop cycle, describing responses in vegetative growth and final grain production. The present study provides a comprehensive evaluation of the mechanism of response of two cultivars of durum wheat to different waterlogging durations through an integrated analysis of an array of physiological, biochemical, biometric, and yield parameters, together with water status, collected at the end of the waterlogging events, during recovery, and at maturity (i.e, BBCH 99, 125 days after waterlogging imposition).

Our results confirmed that a large variation in wheat responses to waterlogging exists, depending on different durations of stress conditions and on diverse genotypic sensitivity [4]. Photosynthesis decreased due to 14 DOW only in Svevo, suggesting a higher waterlogging sensitivity of this cultivar compared with Emilio Lepido, whereas the CO_2_ assimilation rate was similarly impaired between cultivars by the longer 35 DOW. These photosynthetic impairments were clearly due to stomatal limitations (i.e., g_s_ showed the same trends as A), suggesting an isohydric behavior of both cultivars [23], while mesophyll impairments were less evident since C_i_ did not accumulate. The interpretation of a minor occurrence of non-stomatal limitations of photosynthesis was supported by the absence of PSII photodamage (i.e., unchanged F_v_/F_m_), as well as by the slight reduction in qP similarly reported between cultivars and for different waterlogging durations. No waterlogging effects on F_v_/F_m_ (i.e., the most widely used photo-oxidative stress marker [24]) have already been reported in common wheat [17]. The increase of WUE_in_ observed at 35 DOW only in Svevo (as also highlighted by the robust and positive correlation of this parameter with Can2, which strongly discriminated these plants from the others) was interesting and unexpected. Water-use efficiency is largely used in the selection of cultivars with high capacity of adaption and high yield in crop breeding projects [25,26].

Our findings indicate that Svevo likely adopted a better strategy to regulate the use of water in an attempt to cope with the longer waterlogging duration. Actually, a reduction in PSII performance (i.e., reduced Φ_PSII_), together with an activation of the dissipation of the excess excitation energy as heat (i.e., increased qNP), were observed only in Emilio Lepido at 35 DOW (qNP, together with root H_2_O_2_, was positively and strongly correlated with Can1, which discriminated Emilio Lepido plants exposed to WL14, and even more those subjected to WL35, from the others). This also confirms that this cultivar was not able to tolerate oxygen deprivation so long (potentially even less than Svevo at physiological level).

As paradoxical as it may sound, waterlogging often reduces water availability to plants [27]; this process is mainly caused by reduced stomatal conductance due to an increased abscisic acid accumulation [28], and reduced root hydraulic conductance [29]. Leaf RWC of both the investigated cultivars was reduced by 14 DOW, even if Ψ_w_ was never affected by waterlogging treatments. Conversely, leaf RWC was slightly increased by 35 DOW; this was likely due to an osmotic adjustment (i.e., reduced Ψ_π_) adopted by the crop to maintain turgor and cell volume under such detrimental conditions. The importance of osmotic adjustment to improve drought tolerance in plants is notorious [30]; the present study confirms that this process may also deserve more interest in terms of plant responses to waterlogging [31]. Overall, the water status parameters confirmed a differential response of durum wheat to increasing durations of waterlogging. On the contrary, these parameters did not highlight cultivar-specific differences, which were instead markedly pointed out by the biochemical measures.

During waterlogging, factors such as decreases in chlorophyll or other components of the photosynthetic apparatus, as a result of nitrogen deficiency and/or negative feedback from carbohydrate accumulation, have been reported as possible causes of reduced CO_2_ fixation. In some conditions, disturbance to cation homeostasis (e.g., K^+^ and Ca^2+^) and the possible damage of leaves from ROS or phytotoxins (e.g., Fe^2+^ or Mn^2+^) might also contribute to this [4,27,32]. The above-mentioned impairment of the leaf gas exchange was actually in accordance with the overall reduction in photosynthetic pigments (i.e., Chl_TOT_ and Car_TOT_) which play a crucial role in light harvesting for photosynthesis. The degradation of chlorophyll and carotenoids was already reported in plants exposed to waterlogging, e.g., as shown in [33], as well as to other environmental stressors, e.g., as shown in [34,35], signifying that the chloroplast ultrastructure and photosynthetic pigments were impaired. No additional variations in leaf pigment parameters were observed due to waterlogging, indicating that leaf photoprotective mechanisms such as changing Chl a/b ratio and β-car and increasing DEPS levels [36] were not activated. This phenomenon was likely due to the absence of a harsh oxidative pressure induced by waterlogging at leaf level, as suggested by the above-mentioned unchanged F_v_/F_m_, and also confirmed by the lack of accumulation of leaf MDA (one of the major indicators of cell membrane damage) [37]. This appears to be a completely different scenario from the one observed at the root level.

Although it has been largely reported that roots are the plant organs mostly affected by waterlogging [4,27], the present study pioneering demonstrated that increased oxidative pressure and accumulation of H_2_O_2_ occurred in the roots of waterlogged durum wheat. This outcome confirms the importance of evaluating the belowground responses as well to fully elucidate the effects of waterlogging on plants. Increased lipid peroxidation was reported in the roots of both cultivars subjected to 14 DOW, although an accumulation of root H_2_O_2_ occurred only in Emilio Lepido. Although root MDA accumulation was not reported at 35 DOW, a strong accumulation of H_2_O_2_ occurred in the roots of both Emilio Lepido and Svevo subjected to longer periods of waterlogging (root H_2_O_2_ was strongly and positively correlated with Can2, which discriminated Svevo plants exposed to WL35 from the others). Excessive MDA accumulation commonly indicates cell membrane damage, which leads to a series of negative physiological and biochemical events, including reduced photosynthesis [38]. Increased H_2_O_2_ production is one of the hallmarks of the low oxygen stress signal [27,39], as well as of other stress signals [40,41]. The elucidation of these varying responses in terms of lipid peroxidation and H_2_O_2_ accumulation reported between cultivars and waterlogging durations undoubtedly needs and suggests further research (the lack of root MDA increase at 35 DOW was particularly unexpected). However, this phenomenon was likely due to the activation/depression of enzymatic and non-enzymatic antioxidants, adopted by plants to regulate the stress response and signaling [4,42]. Among antioxidants, the key role of phenylpropanoids in the response of durum wheat to waterlogging has been previously indicated by [10].

Variations in lipid peroxidation and H_2_O_2_ accumulation due to genotype and duration also appeared to be linked to specific regulations of membrane transporters, which were investigated at both the leaf and root levels. Membrane transporters are known to play a crucial role in mediating adaptive responses to oxygen deprivation and waterlogging, especially at the root level [27]. Specifically, under such detrimental conditions, root K^+^ uptake is commonly and markedly reduced [43,44], so the ability of roots to maintain cytosolic K^+^ homeostasis and K^+^ channel functionality was named as an essential component of plant acclimation to hypoxia [45]. Conversely, hypoxia commonly induces a rapid elevation in the cytosolic Ca^2+^ concentration in plant cells [27,46]. In addition, under waterlogging, the energy stored in roots can be reduced by inhibiting the active transport of these ions to other organs [38]. The present responses of durum wheat in terms of root K^+^ content were fully in accordance with the above-mentioned reductions in the CO_2_ assimilation rate observed only in Emilio Lepido at 14 DOW and in both cultivars at 35 DOW, whereas leaf K^+^ content decreased in both cultivars, regardless of waterlogging duration (leaf K^+^, together with Car_TOT_, was strongly and negatively correlated with Can1, which discriminated Emilio Lepido plants subjected to WL14 and even more those exposed to WL35 from the others). An elevation in Ca^2+^ content was instead observed only in leaf tissue, specifically in Svevo at 14 DOW and in Emilio Lepido at 35 DOW, indicating that waterlogging disturbed not only the mineral uptake, but also the transport of ions to the aerial organs that might have impaired the stomatal conductance and negatively affected the CO_2_ fixation, translocation, and utilization of assimilates. Our findings corroborate those of [47], which found that stress-induced production of ROS results in anomalies in several important cellular biochemical pathways/reactions. These mechanisms operate in cellular organelles like chloroplast and mitochondria, activating Ca^2+^- and K^+^-permeable cation channels at the plasma membrane. Thereby they also mediate Ca^2+^-based signaling events and K^+^ ion leakage. These outcomes not only confirm the importance of cation homeostasis in waterlogging response but also the higher physiological sensitivity of Svevo reported at 14 DOW and the inability of both cultivars (Emilio Lepido results more sensitive in terms of WUE_in_ and PSII performance) to tolerate the longer oxygen deprivation (i.e., 35 DOW).

The contrasting responses in physiological, water status, and biochemical parameters were only partially confirmed by the biometric measurements. The different reactions between cultivars at 14 DOW were not accordingly highlighted by biomass production, as the number of culms, shoot, and root biomass and the shoot-to-root biomass ratio were similarly affected in both Emilio Lepido and Svevo. In particular, waterlogging induced different biomass distribution regardless of the cultivar. Conversely, number of culms and the amount of shoot biomass indicated a higher sensitivity of Svevo at 35 DOW, suggesting that the strategy adopted by this cultivar in terms of WUE_in_ and the preservation of PSII performance was not successful in terms of biomass production. The root biomass of the two cultivars was similarly impaired by 35 DOW, confirming that root dry weights significantly decrease with waterlogging longer than 20 days [5]. Yet, the growth of roots and leaves are coordinated, and their relative sizes vary dynamically in response to environmental conditions to optimize the utilization of assimilates and other resources [48]. Thus, the increased shoot-to-root biomass ratio of Emilio Lepido exposed to 35 DOW highlighted that, similarly to common wheat [20], the root growth of durum wheat is also inhibited more than shoot growth, as the adventitious root growth could not fully compensate for loss of seminal roots [11].

Although the detrimental effects due to waterlogging events on photosynthesis and PSII performance were no longer detectable at recovery, this phenomenon was not due to an ability of durum wheat to recover its optimal physiological functioning (it is interesting to note that at this time WUE_in_ was reduced in both cultivars previously subjected to 35 DOW); instead it was due to a mismatch between the developmental stages of the control and the waterlogged plants, i.e., the controls were closer to maturity and thus lowered the photosynthetic process. We can thus infer that the plant growth had been slowed down by prolonged water excess, as similarly demonstrated in other winter cereals by [16,17].

The above-mentioned damages could not be recovered and definitively compromised final biomass production and grain yield, as shown by our outcomes for physiological maturity. The grain yield of both cultivars revealed greater reduction with longer waterlogging duration (i.e., 35 DOW), corroborating previous results for waterlogging imposed at tillering in durum wheat that displayed differences yield losses related to waterlogging duration [5]. On the other hand, the same authors [5] also reported a significant reduction in grain yield of the durum wheat cultivars Claudio and Svevo only when waterlogging at tillering was prolonged to more than 20 days. Our present results only partially confirmed those outcomes. This was true only for Emilio Lepido, while Svevo showed a significant decrease in grain yield with both waterlogging durations. The mean temperatures experienced throughout the 35 days of waterlogging were, in this experiment, about 10 °C, whereas they were less than 6 °C in previous research on durum wheat [5]. Thus, higher temperatures during waterlogging can be responsible for the different behavior of Svevo, further confirming that effects on winter cereals can greatly vary due to meteorological conditions.

From an agronomic point of view, plant tolerance to waterlogging involves the maintenance of a relatively high grain yield under waterlogged conditions relative to non-waterlogged conditions. Accordingly, our findings clearly showed that Emilio Lepido was more tolerant to waterlogging, whereas Svevo was more sensitive even with a waterlogging duration shorter than 20 days. To the best of our knowledge, any other durum wheat cultivar from Claudio and Svevo has been investigated for agronomic waterlogging tolerance [5,10]. However, high-yielding genotypes of common wheat were more affected by waterlogging than lower yielding types, because plants were not able to maintain high tillering, as showed by [20]. Our findings corroborated their hypothesis also for durum wheat, because Svevo was more productive in well-drained conditions and had more culms per plant, as compared to Emilio Lepido. Moreover, Svevo has been proved to have higher allocation of biomass in roots during vegetative growth and post-heading dry matter accumulation [49]. The fact that the number of culms and the amount of root biomass in Svevo were more intensely restrained by waterlogging (the number of culms was positively and negatively correlated with Can2, which discriminated Svevo plants exposed to WL35 from the others), further confirmed this hypothesis.

## 4. Materials and Methods

### 4.1. Experimental Site Characteristics

The research was carried out from December 2020 to June 2021 at the field station of the Department of Agriculture, Food and Environment of the University of Pisa, Italy (43°40′ N, 10°19′ E, 1 m a.s.l). The climate of the area is hot in summer Mediterranean conditions (Csa), with mean annual maximum and minimum daily air temperatures of 20.2 and 9.5 °C, respectively, and a mean rainfall of 971 mm per year. The daily air minimum and maximum temperatures and rainfall were recorded throughout the entire period of the research by an automatic meteorological station located close to the experimental site.

### 4.2. Experimental Design and Crop Management

The experimental design consisted of two durum wheat cultivars exposed to 14 and 35 days of waterlogging (DOW) at the tillering stage, compared to well-drained controls. We used the two commercial cultivars Svevo and Emilio Lepido. Svevo is a very early maturing cultivar that was released in 1996 from the genealogy CIMMYT line/Zenit and is high yielding. Emilio Lepido is a more modern cultivar, released in 2011 from the genealogy Orobel/Arcobaleno/Svevo, which matures early and is resistant to cold temperatures. Both have a good resistance to lodging. Plants were grown in 16-L pots made from polyvinyl chloride (PVC) tubes (80 cm long and 16 cm in diameter) fitted with a PVC base. A 30 mm diameter hole was drilled in the bottom of each pot, which was fitted with a 0.9 mm mesh to contain roots and substrate loss. Pots were filled with a sandy-loam soil collected from an adjacent field that was previously cultivated with rapeseed. Main soil properties were: 55.3% sand (2 mm < ∅ < 0.05 mm), 33.8% silt (0.05 mm < ∅ < 0.002 mm), 10.9% clay (<0.002 mm), 7.6 pH, 0.7 g kg^−1^ total nitrogen (Kjeldahl method), 4.5 mg kg^−1^ available P (Olsen method), and 68.9 mg kg^−1^ available K (BaCl_2_-TEA method). The crop was sown on 15 December 2020 within the optimum sowing time for winter cereal production in central Italy. After emergence, the seedlings were thinned to eight plants per pot, corresponding to 400 plants m^–2^. Phosphorus and potassium were applied pre-planting as triple mineral phosphate and potassium sulfate, at the rates of 150 kg ha^−1^ of P_2_O_5_ and K_2_O. Nitrogen was applied at the rate of 150 kg N ha^−1^, and split into three applications at sowing, at pseudo-stem erection (BBCH 30), and at first node detectable (BBCH 31) as urea, in the following proportions: 30–60–60 kg N ha^–1^. The rate of mineral N supply was the recommended value for optimal durum wheat production in central Italy, and the adopted splitting management was proved to be an optimal mineral fertilization practice to ensure both production quantity and quality in the Mediterranean climate [50]. Throughout the experiment, phenological phases were recorded using the BBCH scale for cereals [51] to determine the timing of waterlogging imposition, N applications and harvest. Weed control was performed by hand hoeing, and no pesticide application was needed. The crop was irrigated from flowering to maturity to prevent drought stress, with a total of 200 mm of water applied. Pots were placed outdoors and kept under drained conditions until the plants reached the tillering stage (BBCH 20) on 24 February 2021, when half of the pots were maintained in well-drained conditions (controls), and the other half were exposed to waterlogging by placing the pots into containers (2 × 1 × 1 m) filled with water. A layer of 1 cm of free water was maintained above the soil surface throughout the period of waterlogging to ensure that the soil was completely saturated by water. Three replicate pots were used for all combinations of treatments.

For each cultivar, at waterlogging imposition (0 DOW–24 February 2021) three replicate pots were harvested to determine biomass and physiological characteristics before waterlogging imposition. At the end of each period of waterlogging—that is after two and five weeks (14 and 35 DOW)—all plants of three waterlogged pots and three well-drained pots (controls) were measured for physiological and biochemical parameters (they were performed on the second and third upper and fully expanded leaves). Then three pots per cultivar were moved from the container filled with water to drained conditions. These pots (WL pots to be measured at maturity) were supplied with the scheduled top-dressing N fertilization and kept in drained conditions until plants reached maturity. Control pots received N at the same time of the waterlogged pots. Additional measurements of physiological and water status parameters were carried out during the recovery period, at 70 days after the beginning of waterlogging (i.e., 56 and 35 days after the end of waterlogging, respectively, for waterlogging prolonged 14 and 35 days), to assess the water status and the physiological activities of control and waterlogged plants. At maturity, three waterlogged and three control pots for each cultivar were harvested to assess final biomass and grain yield production.

### 4.3. Plant Measurements

#### 4.3.1. Leaf Gas-Exchange and Chlorophyll *a* Fluorescence

The CO_2_ assimilation rate (A), stomatal conductance (g_s_), and intercellular CO_2_ concentration (C_i_) were determined using a LI-6400 portable photosynthesis system equipped with a 2 × 3 cm chamber and a 6400-02B LED light source (Li-COR Inc., Lincoln, NE, USA), operating at 400 ppm of CO_2_ concentration, 25 ± 2 °C of leaf temperature, 45 ± 5% of RH, 1.8 ± 0.2 kPa of VPD and saturating light conditions (1500 µmol m^−2^ s^−1^ PAR). Intrinsic water-use efficiency (WUE_in_) was calculated as A/g_s_.

After a 40 min dark-adaptation of leaves (same used for leaf gas-exchange measurements), the maximum quantum efficiency of the photosystem II (PSII) photochemistry (F_v_/F_m_), the PSII-operating efficiency in light conditions (Φ_PSII_), the photochemical quenching (qP), and the non-photochemical quenching (qN) were determined by a PAM-2000 chlorophyll *a* fluorometer (Walz, Effeltrich, Germany), set as reported by [52].

#### 4.3.2. Leaf Water Status

Water status parameters were determined at mid-day, according to [53]. Leaf water potential was measured using a Scholander pressure chamber (model 600 Pressure Chamber Instrument, PMS Instrument Company, Albany, NY, USA). Leaf osmotic potential was converted from osmolality (using the Van’t Hoff equation) determined by a VAPRO^®^ Vapor Pressure Osmometer (EliTech Group, Puteaux, France). Relative water content was calculated as (FW-DW)/(TW-DW) × 100, where FW is the fresh weight, TW is the turgid weight after rehydrating samples for 24 h, and DW is the dry weight after oven-drying leaves at 60 °C until constant weight.

#### 4.3.3. Leaf Pigments

Leaf pigments were determined by ultra-high performance liquid chromatography (UHPLC) using a Dionex UltiMate 3000 system equipped with an Acclaim 120 C18 column (5 μm particle size, 4.6 mm internal diameter × 150 mm length) maintained into a Dionex TCC-100 column oven at 30 °C, and a Dionex UVD 170U detector (Thermo Scientific, Waltham, MA, USA; [54]. Leaf material (50 mg fresh weight, FW) was homogenized in 1 mL of 100% HPLC-grade methanol and incubated overnight at 4 °C in the dark. The sample supernatants were filtered through 0.2 μm Minisart^®^ SRT 15 aseptic filters. The pigments were eluted using 100% solvent A (acetonitrile/methanol, 75/25, v/v) for the first 14 min to elute xanthophylls (neoxanthin, Neo; violaxanthin, Vio; antheraxanthin, Ant; lutein, Lut; zeaxanthin, Zea; in order of elution), followed by a 1.5 min linear gradient to 100% solvent B (methanol/ethylacetate, 68/32, v/v), which was pumped for 14.5 min to elute chlorophyll b (Chl b) and chlorophyll a (Chl a) and β-carotene (β-car), followed by 2 min linear gradient to 100% solvent A. The flow rate was 1 mL min^−1^. The column was allowed to re-equilibrate in 100% solvent A for 1 min before the next injection. The pigments were detected by their absorbance at 445 nm. To quantify the pigment content, known amounts (0.003–0.5 mg mL^−1^) of pure standards (Sigma-Aldrich, St. Louis, MO, USA) were injected into the UHPLC system, and an equation correlating the peak area to pigment concentration was formulated. Chromatographic data were processed and recorded by Chromeleon Chromatography Management System software, version 7.2.10–2019 (Thermo Scientific). Total chlorophyll content (Chl_TOT_) was calculated as Chl a + Chl b. Total carotenoid content (Car_TOT_) was calculated as Neo + Vio + Ant + Lut + Zea + β-car, while the xanthophyll cycle pigment content (VAZ) was calculated as Vaz + Ant + Zea. The de-epoxidation state (DEPS) was calculated as (Ant + Zea)/VAZ.

#### 4.3.4. Leaf and Root Lipid Peroxidation and Hydrogen Peroxide

Lipid peroxidation was measured by the thiobarbituric acid reactive substances (TBARS) method, according to [55]. Briefly, 30 mg of leaf samples were extracted with 750 mL of 0.1% trichloroacetic acid (TCA), sonicated three times for 10 min and centrifuged at 13,000× *g* for 10 min at 4 °C. Then, 100 µL of each sample supernatant was mixed with 400 µL of 20% TCA and 0.5% thiobarbituric acid (TBA). Samples were incubated at 95 °C for 30 min and centrifuged at 12,000× *g* for 10 min at 4 °C. The supernatant was measured for absorbances at 440, 532, and 600 nm, using a fluorescence/absorbance microplate reader (Victor3 1420 Multilabel Counter, Perkin Elmer, Waltham, MA, USA). The amount of malondialdehyde (MDA) was calculated as 106 × ((A − B)/157,000), where A = (Abs 532 + TBA − Abs 600 + TBA) − (Abs 532-TBA − Abs 600-TBA) and B = (Abs 440 + TBA − Abs 600 + TBA) × 0.0571.

Hydrogen peroxide content was measured using the Amplex™ Red Hydrogen Peroxide/Peroxidase Assay Kit (Molecular Probes, Life Technologies Corp., Carlsbad, CA, USA), according to [56]. After extraction with potassium phosphate buffer (20 mM, pH 6.5), H_2_O_2_ was determined with the above-reported fluorescence/absorbance microplate reader at 530 and 590 nm for the excitation and emission of resorufin fluorescence, respectively.

#### 4.3.5. Leaf and Root Cations

Leaf and root K^+^ and Ca^2+^ contents were determined by Ion Chromatography (Dionex Aquion, Dionex IonPac™ CS12A, Dionex Cation Self-Regenerating Suppressor CSRS™ 300 4 mm; Sunnyvale, CA, USA). According to [57], 12.5 mg FW of leaf and root tissues were suspended in 4.0 mL of HPLC-grade water, shaken for 15 min and centrifuged at 2100× *g* for 10 min. After filtration through 0.2 μm Minisart^®^ SRT 15 aseptic filters, supernatants were eluted with 20 mM methanesulfonic acid at 1 mL min^−1^.

#### 4.3.6. Crop Growth

At all harvesting times (0, 14, and 35 DOW), subsequently to the above-mentioned physiological measurements, plants were manually cut at ground level. After the shoots were removed, the roots were recovered from the soil by gently washing them with a soft water flow. Washing water was additionally filtered through a fine mesh to prevent root loss. Root sub-samples were used for the biochemical analyses after recording their fresh weight. At maturity (BBCH 99), the shoots were partitioned into culms, leaves, and spikes and spikes separated into kernels, and chaff, and the roots were recovered and measured with the same above-reported procedure.

Biomass of roots, vegetative above-ground plant parts (VAP) and grain yield were determined. For dry weight (DW) determination of all plant parts, the samples were oven-dried at 65 °C to achieve a constant weight.

### 4.4. Statistical Analyses

The Shapiro-Wilk test was used to evaluate the normal distribution of data and homogeneity of variances was tested through Levene’s tests, prior to analyses. The effects of cultivar, waterlogging, and their interaction on the investigated parameters were assessed by a two-way analysis of variance (ANOVA), using Tukey’s test as the post hoc test. Statistically significant effects were considered for *p* ≤ 0.05. Statistical analyses were run in JMP 13.2.0 (SAS Institute Inc., Cary, NC, USA).

A discriminant analysis was applied to the full set of parameters collected at the end of waterlogging treatments to select those that best discriminated among cultivars (Emilio Lepido and Svevo), waterlogging treatment (control and waterlogged), and waterlogging duration (14 and 35 days).

## 5. Conclusions

In conclusion, our pioneering study demonstrated that waterlogging imposed to durum wheat at tillering (i) impaired photosynthetic activity, mainly due to stomatal limitations, pigment degradations and altered cation homeostasis; (ii) determined oxidative damage and H_2_O_2_ accumulation in the root systems; and (iii) finally depressed the grain yield, due to slowed-down tiller formation and crop growth. Additionally, our results showed that genotypic differences in waterlogging tolerance of durum wheat are present. The two genotypes differed not only in their immediate responses to waterlogging but also in the recovery of growth once the soil was drained. Consequently, the final grain yields of the two cultivars were affected differently. As a matter of fact, one cultivar (Emilio Lepido) was more tolerant to waterlogging than the other (Svevo).

Therefore, our results suggest that the waterlogging tolerance of durum wheat can be achieved by pyramiding the numerous physiological, water status, and biochemical parameters that confer efficient key processes such as energy maintenance, cytosolic ion homeostasis, and ROS control and detoxification, and consequently ensure satisfying biomass production and yield.

Additional research is obviously required to evaluate the responses of present and other cultivars of durum wheat to waterlogging, under several environmental and cropping conditions. This would give a better picture of overall crop performance and allow further endorsement of our present results.

## Figures and Tables

**Figure 1 plants-10-02357-f001:**
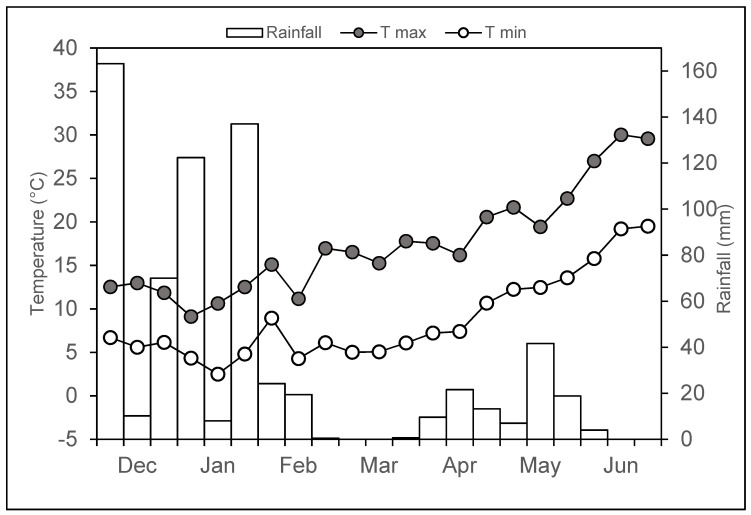
Air minimum (white dots) and maximum (black dots) temperatures and rainfall (bars) during the cropping season (December 2020–June 2021).

**Figure 2 plants-10-02357-f002:**
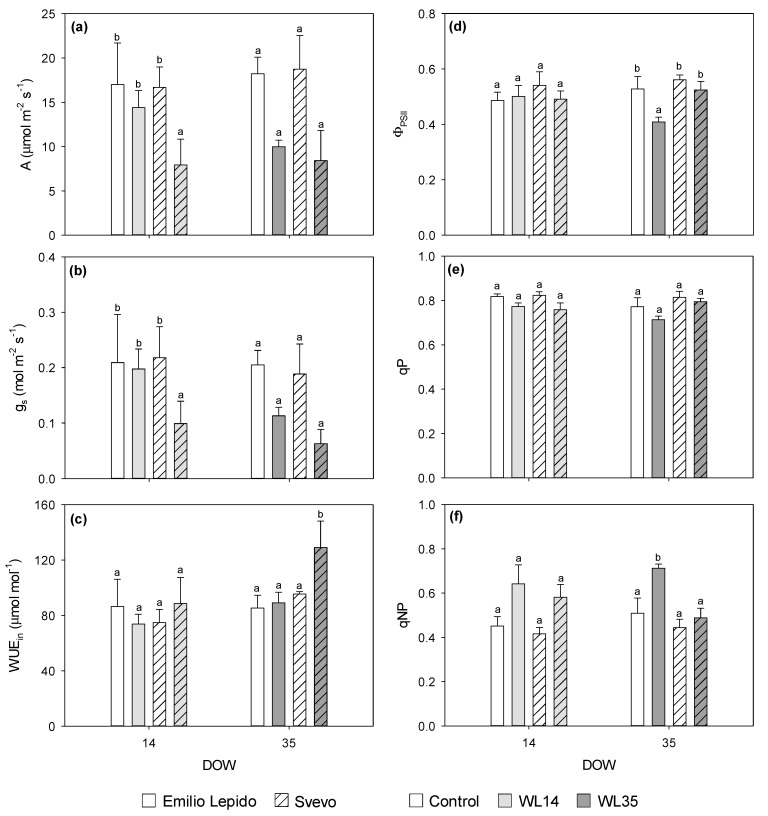
(**a**) CO_2_ assimilation rate (A), (**b**) stomatal conductance (g_s_), (**c**) intrinsic water-use efficiency (WUE_in_), (**d**) PSII-operating efficiency in light conditions (Φ_PSII_), (**e**) photochemical quenching (qP), and (**f**) non-photochemical quenching (qNP) in the durum wheat cultivars Emilio Lepido (solid) and Svevo (pattern) subjected to 0 (i.e., control; white), 14 (i.e., WL14; light gray), or 35 (i.e., WL35; dark gray) days of waterlogging (DOW). Data are mean ± standard deviation. For each waterlogging duration, according to Tukey’s post hoc test (*p* ≤ 0.05), different letters indicate significant differences among means (*p* ≤ 0.05).

**Figure 3 plants-10-02357-f003:**
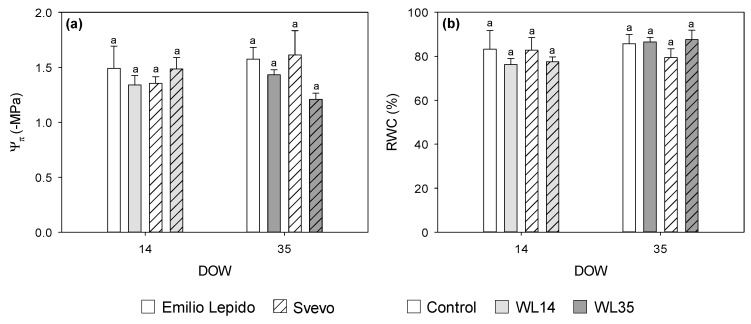
(**a**) Leaf osmotic potential (Ψ_π_), and (**b**) leaf relative water content (RWC) of the durum wheat cultivars Emilio Lepido (solid) and Svevo (pattern) subjected to 0 (i.e., control; white), 14 (i.e., WL14; light gray), or 35 (i.e., WL35; dark gray) days of waterlogging (DOW). Data are mean ± standard deviation. For each waterlogging duration, according to Tukey’s post hoc test (*p* ≤ 0.05), different letters indicate significant differences among means (*p* ≤ 0.05).

**Figure 4 plants-10-02357-f004:**
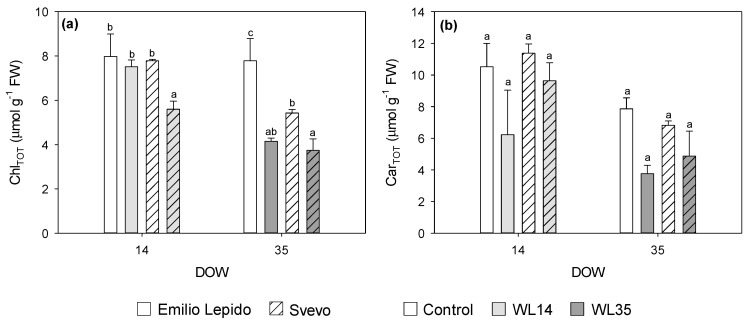
(**a**) Total chlorophyll (Chl_TOT_), and (**b**) total carotenoid (Car_TOT_) content in leaves of the durum wheat cultivars Emilio Lepido (solid) and Svevo (pattern) subjected to 0 (i.e., control; white), 14 (i.e., WL14; light gray), or 35 (i.e., WL35; dark gray) days of waterlogging (DOW). Data are mean ± standard deviation. FW: fresh weight. For each waterlogging duration, according to Tukey’s post hoc test (*p* ≤ 0.05), different letters indicate significant differences among means (*p* ≤ 0.05).

**Figure 5 plants-10-02357-f005:**
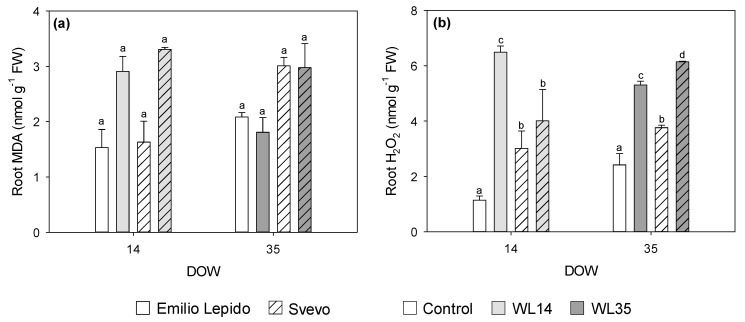
(**a**) Malondialdehyde (MDA) and (**b**) hydrogen peroxide (H_2_O_2_) content in the roots of the durum wheat cultivars Emilio Lepido (solid) and Svevo (pattern) subjected to 0 (i.e., control; white), 14 (i.e., WL14; light gray), or 35 (i.e., WL35; dark gray) days of waterlogging (DOW). Data are mean ± standard deviation. FW: fresh weight. For each waterlogging duration, according to Tukey’s post hoc test (*p* ≤ 0.05), different letters indicate significant differences among means (*p* ≤ 0.05).

**Figure 6 plants-10-02357-f006:**
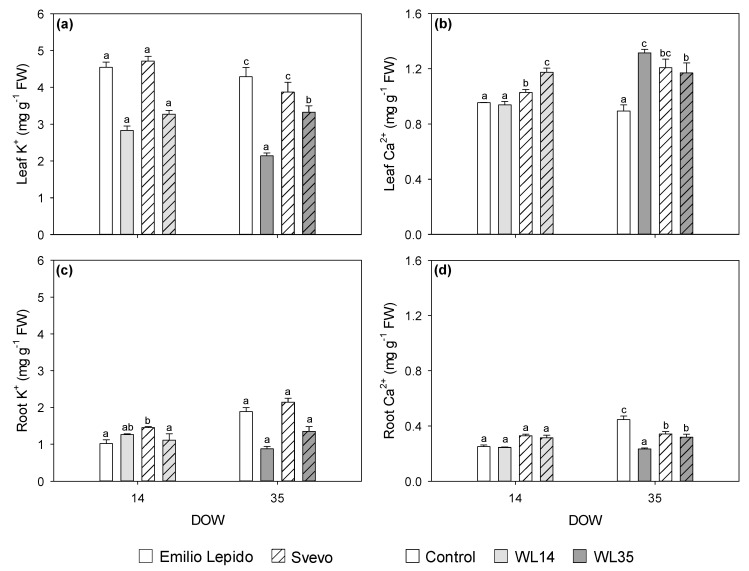
(**a**) Leaf K^+^ (**b**), leaf Ca^2+^ (**c**), root K^+^, and (**d**) root Ca^2+^ content of the durum wheat cultivars Emilio Lepido (solid) and Svevo (pattern) subjected to 0 (i.e., control; white), 14 (i.e., WL14; light gray), or 35 (i.e., WL35; dark gray) days of waterlogging (DOW). Data are mean ± standard deviation. FW: fresh weight. For each waterlogging duration, according to Tukey’s post hoc test (*p* ≤ 0.05), different letters indicate significant differences among means (*p* ≤ 0.05).

**Figure 7 plants-10-02357-f007:**
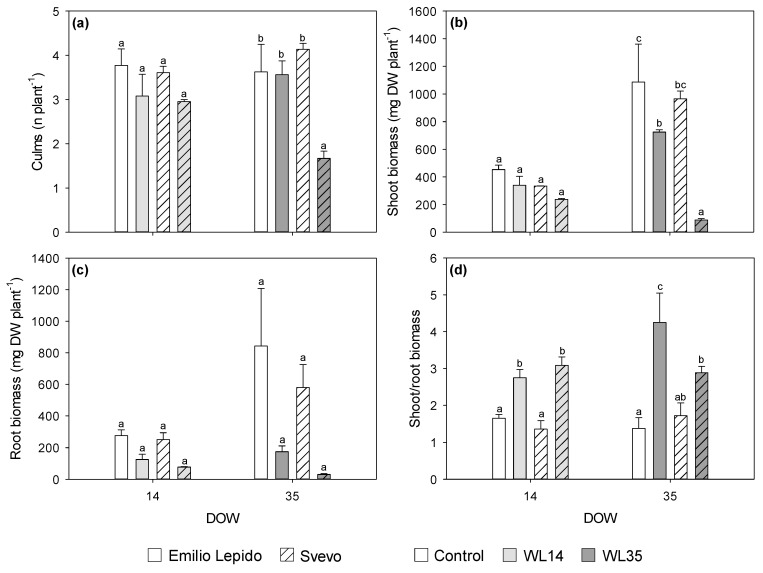
(**a**) Number of culms per plant (**b**), shoot biomass (**c**), root biomass, and (**d**) shoot-to-root ratio of the durum wheat cultivars Emilio Lepido (solid) and Svevo (pattern) subjected to 0 (i.e., control; white), 14 (i.e., WL14; light gray), or 35 (i.e., WL35; dark gray) days of waterlogging (DOW). Data are mean ± standard deviation. DW: dry weight. For each waterlogging duration, according to Tukey’s post hoc test (*p* ≤ 0.05), different letters indicate significant differences among means (*p* ≤ 0.05).

**Figure 8 plants-10-02357-f008:**
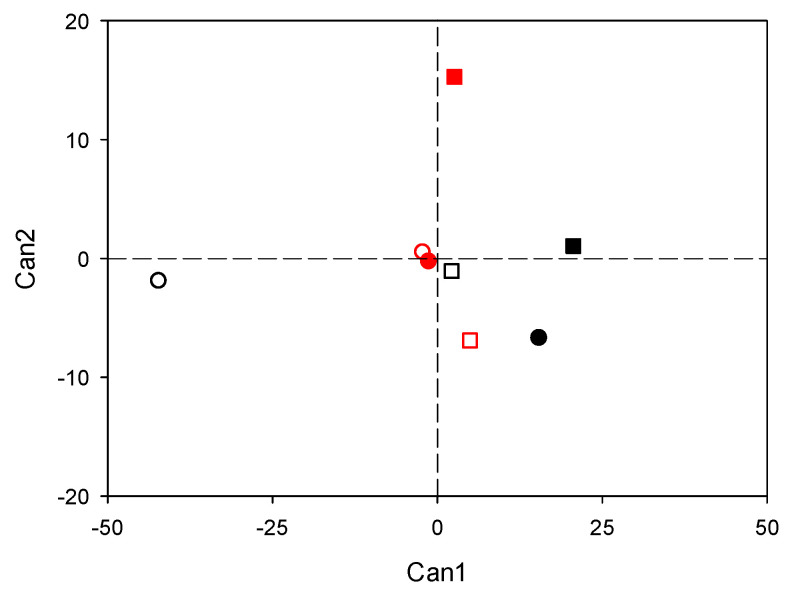
Discrimination between cultivar (Emilio Lepido, black; Svevo, red), waterlogging treatment (control, open; waterlogged, closed), and waterlogging duration (14 days, circle; 35 days, square) on the basis of canonical discriminant analysis applied to the full set of parameters collected at the end of waterlogging treatments. The first two canonicals are shown (Can1 and Can2).

**Table 1 plants-10-02357-t001:** Results of two-way analysis of variance (ANOVA) for the effects of cultivar (*C*; degrees of freedom, df: 1), waterlogging (*WL*; df: 1), and their interaction (*C × WL*; df: 1) on water status, and physiological, biochemical, and biometric parameters in durum wheat cultivars Emilio Lepido and Svevo subjected to 0, 14, or 35 days of waterlogging (DOW). Data are F values and *p* levels (***: *p* ≤ 0.001, **: *p* ≤ 0.01, *: *p* ≤ 0.05, ns: *p* > 0.05). ND: not determinable (i.e., all plants with one culm per plant).

Parameter	0 DOW			14 DOW			35 DOW		
	*C*	*WL*	*C × WL*	*C*	*WL*	*C × WL*	*C*	*WL*	*C × WL*
A	0.26 ns	0.61 ns	0.04 ns	7.05 *	19.74 ***	5.78 *	0.15 ns	45.83 ***	0.56 ns
g_s_	0.51 ns	0.01 ns	0.02 ns	3.54 ns	7.52 *	5.10 *	0.07 ns	42.04 ***	0.99 ns
C_i_	4.38 ns	0.07 ns	0.00 ns	1.05 ns	0.46 ns	0.01 ns	3.18 ns	1.56 ns	0.20 ns
WUE_in_	0.73 ns	0.11 ns	0.08 ns	0.07 ns	0.01 ns	4.78 *	19.86 ***	11.03 **	7.03 *
F_v_/F_m_	21.59 ***	0.02 ns	0.07 ns	0.03 ns	3.34 ns	0.04 ns	0.06 ns	1.19 ns	0.14 ns
Φ_PSII_	0.26 ns	0.75 ns	0.69 ns	1.77 ns	1.07 ns	3.62 ns	24.77 ***	26.91 ***	7.62 *
qP	0.70 ns	0.07 ns	0.15 ns	0.23 ns	37.78 ***	1.42 ns	21.51 ***	9.10 *	2.11 ns
qNP	0.20 ns	0.25 ns	0.17 ns	3.43 ns	47.63 ***	0.26 ns	39.99 ***	29.36 ***	12.16 **
Ψ_w_	2.40 ns	0.27 ns	0.27 ns	0.03 *	3.98 ns	0.05 ns	1.50 ns	1.50 ns	1.50 ns
Ψ_π_	0.40 ns	0.42 ns	0.00 ns	0.01 ns	0.02 ns	3.05 ns	2.12 ns	18.41 **	4.18 ns
RWC	6.62 *	0.00 ns	0.15 ns	0.03 ns	5.11 *	0.11 ns	1.94 ns	5.74 *	3.75 ns
Chl_TOT_	11.90 **	0.06 ns	0.28 ns	10.52 *	16.58 **	7.06 *	17.54 **	64.52 ***	8.65 *
Car_TOT_	2.84 ns	0.05 ns	0.01 ns	4.64 ns	9.26 *	1.66 ns	0.00 ns	32.77 ***	4.20 ns
Chl a/b	149.67 ***	0.20 ns	0.03 ns	24.29 **	3.47 ns	4.27 ns	2.17 ns	1.92 ns	0.61 ns
β-car	4.52 ns	0.02 ns	0.00 ns	16.02 **	1.87 ns	0.45 ns	1.50 ns	1.98 ns	0.64 ns
DEPS	647.98 ***	0.01 ns	0.14 ns	2.45 ns	2.90 ns	3.99 ns	1.80 ns	3.24 ns	0.80 ns
Leaf MDA	139.95 ***	0.00 ns	0.12 ns	2.25 ns	24.71 **	1.08 ns	63.41 ***	16.66 **	4.08 ns
Leaf H_2_O_2_	329.04 ***	0.06 ns	0.00 ns	0.76 ns	2.38 ns	1.39 ns	369.70 ***	14.28 **	2.41 ns
Root MDA	883.67 ***	0.77 ns	0.43 ns	3.06 ns	114.77 ***	1.13 ns	60.53 ***	1.32 ns	0.82 ns
Root H_2_O_2_	5.68 *	0.75 ns	0.45 ns	0.86 ns	91.29 ***	42.98 ***	97.88 ***	569.87 ***	5.39 *
Leaf K^+^	8.31 *	0.54 ns	1.00 ns	17.71 **	473.36 ***	3.66 ns	10.36 *	127.35 ***	44.91 ***
Leaf Ca^2+^	29.00 ***	1.27 ns	0.72 ns	148.98 ***	26.72 ***	41.21 ***	7.09 *	37.48 ***	53.49 ***
Root K^+^	1.50 ns	0.00 ns	0.68 ns	6.12 *	0.64 ns	25.33 **	34.83 ***	214.65 ***	3.13 ns
Root Ca^2+^	57.15 ***	0.62 ns	0.01 ns	99.30 ***	1.88 ns	0.47 ns	0.81 ns	108.80 ***	69.69 ***
Culms	ND	ND	ND	0.62 ns	13.41 **	0.01 ns	10.76 *	35.98 ***	32.51 ***
Shoot biomass	25.86 ***	0.00 ns	0.00 ns	27.39 ***	24.53 **	0.16 ns	21.77 **	58.30 ***	10.09 *
Root biomass	27.08 ***	0.00 ns	0.00 ns	3.71 ns	73.85 ***	0.39 ns	3.18 ns	28.57 ***	0.27 ns
Shoot-to-root biomass	18.88 **	0.00 ns	0.00 ns	0.86 ns	148.43 ***	7.43 *	0.097 ns	55.97 ***	10.04 *

Parameter abbreviations: A, CO_2_ assimilation rate; g_s_, stomatal conductance; C_i_, intercellular CO_2_ concentration; WUE_in_, intrinsic water-use efficiency (i.e., A/g_s_); F_v_/F_m_, maximum quantum efficiency of the photosystem II (PSII) photochemistry; Φ_PSII_, PSII-operating efficiency in light conditions; qP, photochemical quenching; qNP, non-photochemical quenching; Ψ_w_, leaf water potential; Ψ_π_, leaf osmotic potential; RWC, relative water content; Chl_TOT_, total chlorophylls; Car_TOT_, total carotenoids; Chl a/b, chlorophyll a/b ratio; β-car, β –carotene; DEPS, de-epoxidation state; MDA, malondialdehyde; H_2_O_2_, hydrogen peroxide; K^+^, potassium ion; Ca^2+^, calcium ion.

**Table 2 plants-10-02357-t002:** Physiological parameters of the durum wheat cultivars Emilio Lepido and Svevo at recovery (70 days from the beginning of waterlogging) and previously subjected to 0, 14, or 35 days of waterlogging (C, WL14, and WL35, respectively). F values and *p* levels (***: *p* ≤ 0.001, **: *p* ≤ 0.01, *: *p* ≤ 0.05, ns: *p* > 0.05) of the two-way analysis of variance (ANOVA) for the effects of cultivar (*C*; degrees of freedom, df: 1), waterlogging (*WL*; df: 2) and their interaction (*C × WL*; df: 2) are shown. In case two-way ANOVA reveals a significant *C* × *WL* interactive effect on the specific parameter, according to Tukey’s post hoc test, different letters indicate significant differences among means (*p* ≤ 0.05).

Parameter	Emilio Lepido			Svevo			*ANOVA*		
	*C*	WL14	WL35	*C*	WL14	WL35	*C*	*WL*	*C × WL*
A	9.3 ± 0.8 ^a^	14.2 ± 1.4 ^c^	9.7 ± 0.1 ^a^	9.9 ± 0.7 ^ab^	12.5 ± 0.2 ^bc^	13.2 ± 2.3 ^c^	2.75 ns	19.83 ***	9.58 ***
g_s_	0.13 ± 0.02	0.18 ± 0.02	0.18 ± 0.04	0.16 ± 0.03	0.20 ± 0.00	0.24 ± 0.03	13.50 **	14.29 ***	1.60 ns
C_i_	262 ± 2	251 ± 5	285 ± 19	275 ± 23	278 ± 1	289 ± 6	7.88 *	6.88 **	1.73 ns
WUE_in_	73 ± 3	77 ± 1	57 ± 13	64 ± 15	62 ± 1	55 ± 3	6.70 ns	6.10 **	1.30 ns
F_v_/F_m_	0.78 ± 0.01	0.79 ± 0.01	0.80 ± 0.00	0.78 ± 0.00	0.79 ± 0.02	0.78 ± 0.01	2.02 ns	0.77 ns	1.77 ns
Φ_PSII_	0.56 ± 0.01 ^a^	0.56 ± 0.03 ^a^	0.64 ± 0.01 ^c^	0.59 ± 0.01 ^ab^	0.60 ± 0.02 ^bc^	0.62 ± 0.03 ^bc^	4.23 ns	21.19 ***	7.50 **
qP	0.80 ± 0.01 ^ab^	0.80 ± 0.02 ^a^	0.89 ± 0.01 ^c^	0.85 ± 0.00 ^bc^	0.85 ± 0.01 ^bc^	0.87 ± 0.05 ^c^	8.41 **	17.95 ***	6.91 **
qNP	0.46 ± 0.00 ^b^	0.46 ± 0.06 ^b^	0.34 ± 0.02 ^a^	0.46 ± 0.01 ^b^	0.35 ± 0.04 ^a^	0.29 ± 0.03 ^a^	13.39 **	34.38 ***	5.56 *

**Table 3 plants-10-02357-t003:** Number of culms and spikes per plant (n plant^−1^), grain yield (g plant^−1^), and the vegetative above-ground part (VAP) and root biomass (g dry weight plant^−1^) of the durum wheat cultivars Emilio Lepido and Svevo at maturity (125 days from the beginning of waterlogging), and previously subjected to 0, 14, or 35 days of waterlogging (C, WL14 and WL35, respectively). F values and *p* levels (***: *p* ≤ 0.001, **: *p* ≤ 0.01, *: *p* ≤ 0.05, ns: *p* > 0.05) of the two-way analysis of variance (ANOVA) for the effects of cultivar (*C*; degrees of freedom, df: 1), waterlogging (*WL*; df: 2), and their interaction (*C × WL*; df: 2) are shown. In case two-way ANOVA reveals a significant *C* × *WL* interactive effect on the specific parameter, according to Tukey’s post hoc test, different letters indicate significant differences among means (*p* ≤ 0.05).

Parameter	Emilio Lepido			Svevo			*ANOVA*		
	*C*	WL14	WL35	*C*	WL14	WL35	*C*	*WL*	*C × WL*
Culms	3.2 ± 0.3	3.3 ± 0.3	2.4 ± 0.2	4.0 ± 1.2	3.1 ± 0.3	2.6 ± 0.1	0.34 ns	0.02 *	1.10 ns
Spikes	2.6 ± 0.0	2.3 ± 0.5	1.8 ± 0.2	2.3 ± 0.9	1.9 ± 0.1	2.0 ± 0.4	0.48 ns	2.61 ns	0.87 ns
Grain yield	2.5 ± 0.1 ^bc^	2.2 ± 0.7 ^b^	1.7 ± 0.2 ^ab^	3.3 ± 0.3 ^c^	1.8 ± 0.2 ^ab^	1.2 ± 0.2 ^a^	0.20 ns	27.45 ***	6.24 *
VAP biomass	5.1 ± 0.3	3.9 ± 1.1	3.4 ± 0.3	5.6 ± 0.6	3.4 ± 0.5	2.6 ± 0.1	0.39 ns	27.70 ***	2.26 ns
Root biomass	1.17 ± 0.14	1.05 ± 0.31	0.78 ± 0.01	0.93 ± 0.24	0.60 ± 0.10	0.54 ± 0.25	10.59 **	5.73 *	0.55 ns

## Data Availability

The data presented in this study are available on request from the corresponding author.

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
