# Peer review of "Transient Waterlogging Events Impair Shoot and Root Physiology and Reduce Grain Yield of Durum Wheat Cultivars"

_plants, 2021, doi:10.3390/plants10112357_

Round 1
Reviewer 1 Report
The MS entitled “Transient waterlogging events impair shoot and root physiology and reduce grain yield of durum wheat cultivars” with authors Lorenzo Cotrozzi, Giacomo Lorenzini, Cristina Nali, Claudia Pisuttu, Silvia Pampana, Elisa Pellegriniis presents valuable new data and is interesting but needs serious improvement to be published.
Climate change is predicted to cause more frequent waterlogging events in the Mediterranean region. That challenge is worth to be studied in detail especially for a staple crop as durum wheat, which is essential for the region. The authors' focus is on root and shoots physiological parameters, including grain yield, an important characteristic of durum wheat. The waterlogging stress was applied for 14 or 35 days, for two different cultivars. Some physiological parameters were evaluated after the recovery period as well. Over 50% of the cited literature are from the last 10 years (36 from 58 references are after the year 2011). However, the manuscript quality will be improved if authors use shorter sentences because they are difficult to understand. It is essential to improve data presentation. It is a journal requirement to add an additional partition “Conclusion” at the end of the MS.
It is very important to make the text above the tables unambiguous, for example, it is not clear if parameters shown in Table 2 are measured after the period of recovery or not (the period is necessary to be mentioned). The following sentence (from Table 2) is not lucid: ”Variation in number of culms and spikes (n plant-1), grain yield (g plant-1), and vegetative above-ground part (VAP) and root biomass (g dry weight plant-1) of wheat cultivars Emilio Lepido and Svevo at maturity (the time gap between the beginning of treatment and maturity is not clear, please include), and previously subjected to 0, 14 or 35 days of waterlogging (C, WL14 and WL35, respectively).”. In addition, there are no statistics' letters for at least one panel per figure. See Fig. 2 (d,e,f), Fig. 4b, Fig. 5a, Fig 6 (a, c, d), Fig. 7 (a-c), and add statistics’ letters. It will be better to move the position of each figure or table after the appropriate paragraph from the Result section (where it was described). Last but not least, authors present some results as percentage average for both cultivars. I strongly recommend presenting the percentage values separately for each cultivar.
English corrections:
L21-26: The sentence should be split up into 2/3 shorter sentences which will ease the reader.
L50-52: Please, correct. The sentence is difficult to understand.
L84: “maybe “ needs correction
L234-236: “The large variation in response of wheat to waterlogging on the basis of waterlogging events of different durations and possible genotypic-specific sensitivity to waterlogging [4] was confirmed by the present study, already by our physiological outcomes.”. - The sentence needs correction in terms of clarity.
L245-247: ”Waterlogging effects on Fv/Fm, the most widely used photo-oxidative stress marker [22], were not previously reported neither in common wheat by [17].” – Needs correction as above.
L248-252: “This parameter, largely used in selection of cultivars with high capacity of adaption and high yield in crop breeding projects [23, 24], indicated that Svevo likely adopted a better strategy to regulate the use of water in attempt to cope with the longer waterlogging duration.” – Please, split the sentence in two.
L260-262 “Conversely, leaf RWC resulted slightly increased by 35 DOW, thanks to an osmotic adjustment (i.e., reduced Ψπ) adopted by the crop to maintain turgor and cell volume under such detrimental conditions.” – Please, correct in terms of clarity.
L265-268: “Overall, the water status parameters confirmed the variation in response of wheat to waterlogging based on different durations of these harmful events, while they did not highlight cultivar-specific differences, which were instead markedly pointed out by biochemical parameters.” - Not clear sentence, please correct.
“Actually, although Ψw was never affected by waterlogging treatments, leaf RWC of both the investigated cultivars was reduced by 14 DOW.” – It will be better to refrain from using two words like actually/although together.
L303 ”…due the activation…” – need correction
L305-306: “Among antioxidants [10] have indicated the key role of phenylpropanoids in the response of durum wheat to waterlogging.” – as above.
L334: “These differential responses highlighted…” – it is not clear which are these responses. Please, correct.
Please refrain from using words as “anyway” (L342)
L380-382 “However, in common wheat [49] showed that high yielding genotypes were more affected by waterlogging than lower yielding types, because they were not able to maintain high tillering.” - Please correct
Other
Please, explain abbreviations at first mentioning. For example, on L107-109: RWC, ChlTOT, MDA, DEPS, etc.
L117/L120-121: ''... slightly reduced by 14 DOW (-7%) and slightly increased by 35 DOW (+5%; Figure 3b)." Please, refrain to explain each slight difference, especially if it is not statistically significant. The same is valid for L158-159.
L148: “Shoot/root biomass ratio…”. I would suggest changing with “Shoot to root biomass ratio”.
L158: “WL35, +35%, as average”. The percentage values for both cultivars should be presented separately, not as average. Please, correct everywhere.
L171: “…whereas the root biomass was reduced (-37%) only by WL35.”. Please, add the cultivar for which the root biomass inhibition (in %) was calculated and include this information everywhere if it was not mentioned.
L233: “physiological maturity”: please give more information about physiological maturity
L326: “…in several important cellular biochemical pathways/reactions…” – please, revise and split the sentence in two.
L339: “…different partitioning in biomass allocation…” – please, change partitioning or allocation with a more appropriate word
L349: ” adventitious root growth” – did the authors make some experiments confirming changes in adventitious root growth, or the suggestion was made by other researchers?
L367: “…conversely…” – please, change with a more appropriate word. The sentence will sound better if it was split into 2.
L413-425: This part should be partitioned between the section Introduction and Discussion.
L469-461: “Before WL imposition and at the end of each period of waterlogging (0, 14 and 35 DOW, respectively) the following physiological determinations were carried out in both WL and C plants.” - Please add the parameters which were measured. It is necessary to describe how you performed the root biomass (as well as other root parameters) measurement because the plants were soil grown.
Author Response
Thank you for revising our paper.

Reviewer 2 Report
The graphical presentation of the research results needs to be improved. The data presented in Figures 2-7 are redundant as these values ​​are summarized in Table 1.Author Response
Thank you for revising our paper.

Reviewer 3 Report
The problem presented in the article is relevant. The authors investigate the biochemical and physiological mechanisms that are caused by flooding and waterlogging of two varieties of durum wheat. The results demonstrated how this stress affects various photosynthetic parameters, tillering, etc. However, there are 2 main remarks about the design of the experiment.
Firstly, this is a small number of varieties - only 2, and there is no information about the flowering time, the length of the vegetation period and etc.. Since the genotype of the variety can significantly affect the studied parameters, the results obtained may depend not on the degree of waterlogging, but on the stage of plant development. May be it is necessary to compose two groups, contrasting in tolerance to waterlogging, but similar in ripening time. At second, the experiments was carried out only in the conditions of the 2020-2021, it must be repeated at least one more season, since other environmental conditions can affect the manifestation of physiological parameters, plant growth and productivity. Or perform a laboratory experiment (if it is possible) under controlled conditions with a sufficient number of repetitions.
I recommend the authors to repeat the experiment under another environments in order to confirm obtained results, and resubmit the article.
Author Response
Thank you for revising our paper.

Round 2
Reviewer 1 Report
The manuscript is improved a lot. However, I respectfully disagree that it is not necessary to show statistics' letters for each figure panel (lack of significance could be still presented by equal letters) and I recommend Major revision. The comments are highlighted in the pdf-file attached.

Reviewer 3 Report
I am ready to agree with the author's response that only two durum wheat cultivars can be used for the studies. The authors showed that both varieties have approximately similar development parameters according to the Zadoks scale. Therefore, the results obtained do not depend on the stage of plant development.
My second comment concerns the need to conduct experiments over several field seasons. In a previously article published by the same authors (Pampana et al., Cereal Research Communication, 2016), it was shown that the growing season significantly affects grain, and straw biomass. Therefore, environmental factors can also affect physiological and biochemical parameters. The results presented in this article on the effect of waterlogging on physiological parameters(for example, photosynthesis) are preliminary and must be confirmed after additional checks for at least 1-2 seasons. Or there must be strong evidence (and literature data on the absence of the influence of environments on physiological parameters.
Round 3
Reviewer 1 Report
Dear authors,
There are still some faults, which are obligatory to be corrected. Please see the highlights and comments within the pdf file (version 3 of the MS).

Reviewer 3 Report
The authors have provided agrumented response to my second point. Also they presented some published results obtained on the basis of only one environments. Unfortunately, these references are quite old (10-15 years old).
However, based on the fact that the effects of waterlogging for durum wheat have not been studied in detail, I suppose that some experimental data in this paper have novelty and can be published. Nevertheless, I strongly recommended the authors to repeat obtained results in other environments.
